



# Source backtracking for dust storm emission inversion using adjoint method: case study of northeast China

Jianbing Jin[1,2], Arjo Segers[3], Hong Liao[1], Arnold Heemink[2], Richard Kranenburg[3], and Hai Xiang Lin[2]

[1]Jiangsu Key Laboratory of Atmospheric Environment Monitoring and Pollution Control, Collaborative Innovation Center of Atmospheric Environment and Equipment Technology, School of Environmental Science and Engineering, Nanjing University of Information Science and Technology, Nanjing, China
[2]Delft Institute of Applied Mathematics, Delft University of Technology, Delft, the Netherlands
[3]TNO, Department of Climate, Air and Sustainability, Utrecht, the Netherlands

**Correspondence:** Jianbing Jin (jianbing.jin@nuist.edu.cn)

**Abstract.** Emission inversion using data assimilation fundamentally relies on having the correct assumptions on the emission background error covariance. A perfect covariance accounts for the uncertainty based on prior knowledge, and is able to explain differences between model simulations and observations. In practice, emission uncertainties are constructed empirically, hence a partially unrepresentative covariance is unavoidable. Concerning its complex parameterization, dust emissions are a typical

example where the uncertainty could be induced from many underlying inputs, e.g., information on soil composition and moisture, landcover and erosive wind velocity, and these can hardly be taken into account together. This paper describes how an adjoint model can be used to detect errors in the emission uncertainty assumptions. This adjoint based sensitivity method could serve as a supplement of a data assimilation inverse modeling system to trace back the error sources, in case that large observation-minus-simulation residues remain after assimilation based on empirical background covariance.

The method follows on application of a data assimilation emission inversion for an extreme severe dust storm over East Asia (Jin et al., 2019b). The assimilation system successfully resolved observation-minus-simulation errors using satellite AOD observations in most of the dust-affected regions. However, a large underestimation of dust in northeast China remained despite the fact the assimilated measurements indicated severe dust plumes there. An adjoint implementation of our dust simulation model is then used to detect the most likely source region for these unresolved dust loads. The backward modeling

points to the Horqin desert as source region, which was indicated as a non-source region by the existing emission scheme. The reference emission and uncertainty are then reconstructed over the Horqin desert by assuming higher surface erodibility. After the emission reconstruction, the emission inversion is performed again and the posterior dust simulations and reality are now in much closer harmony. Based on our results, it is advised that emission sources in dust transport models include Horqin desert as a more active source region.

## 1 Introduction

Severe dust storms are relatively common events in arid or semi-arid regions over the globe, e.g., in North Africa, the Middle East, Southwest Asia and East Asia, and Australia (Shao et al., 2013). Dust particles could be lifted several kilometers high into



the atmosphere, subsequently carried over distances of thousands kilometers by the prevailing winds. Substantial amounts of dust particles in dust storms are a great threat to human health and properties in areas downwind of dust source regions (World Meteorological Organization, 2018; Basart et al., 2019). The impact on human health consists of dust pneumonia, strep throat, cardiovascular disorders and eye infections. Dust storms can also carry irritating spores, bacteria, viruses and persistent organic

pollutants (World Meteorological Organization, 2017). Next to the human health, the resulting low visibility can cause a severe disruption of the transportation system. For instance, struck by a choking dust storm, the visibility in Beijing has plummeted and over 1,100 flights were delayed in early May 2017 (Jin et al., 2019b). The dust cycle itself is also a key player in the Earth system with profound effects on terrestrial and ocean fertilization, precipitation (Benedetti et al., 2014) and atmospheric radiation (Kosmopoulos et al., 2017).

Due to the growing interest in dust storms, the understanding of the physical processes associated with the dust cycles has increased rapidly over the last decades (World Meteorological Organization, 2018). To improve the simulation skill of dust models, many studies were carried out to parameterize the emission rates using wind tunnel tests or field experiments (Shao et al., 1996; Marticorena and Bergametti, 1995; Alfaro et al., 1998; Fécan et al., 1999). These emission parameterization schemes were then incorporated into large-scale global chemical transport models, e.g., CAMS-ECMWF (Morcrette et al.,

2009), or regional ones, e.g., NASA-GEOS-5 (Colarco et al., 2010) and BSC-DREAM8b (Basart and Carlos, 2012). An important application of these models is to forecast dust concentrations over a few hours to a few days in order to reduce the potential impact on society (Wang et al., 2000; Gong et al., 2003). Different from anthropogenic aerosols, dust particles arise from a complex erosion process with extremely high spatial and temporal variability. A crucial element for the correct simulation (and forecast) of dust transport is the correct representation of the source areas and emission rates. In large-scale

modeling systems, this representation remains relatively crude, due to uncertainty in the different input data such as soil properties (most important soil texture data), surface roughness, landcover (vegetation), topography, as well as insufficient knowledge about the aerosol lifting process itself (Escribano et al., 2016). Besides, quality of forecast of relatively coarse resolution models for wind fields and soil moisture can impact prognostic quality of dust emission and transport. The difficult task to describe all of these inputs correctly subsequently leads to nontrivial simulation errors. Large discrepancies (a factor

up to 10) in dust emissions among models were reported in the evaluation of multiple models participating in the Aerosol Comparison between Observations and Models (AeroCom) phase I experiments (Huneeus et al., 2011; Koffi et al., 2012); the observation-minus-simulation difference can even be as large as two orders of magnitudes (Uno et al., 2006; Gong and Zhang, 2008).

Recent advances in sensor technology and the reduced cost of monitoring systems have led to an increase of observation data

that could be used to analyze dust storms. These observations could be used to explore and improve aerosol emission modeling through inverse modeling. Progress in dust emission inversion has been made in the last decade by assimilating column-integrated satellite aerosol properties (e.g., Moderate Resolution Imaging Spectroradiometer (MODIS) in Schutgens et al. (2012); Khade et al. (2013); Yumimoto and Takemura (2015); Yumimoto et al. (2016a); Di Tomaso et al. (2017), Himawari-8 in Jin et al. (2019b)), Cloud-Aerosol LIdar with Orthogonal Polarization (CALIPSO) vertical aerosol profiles in Sekiyama

et al. (2010), and ground-based $PM_{10}$ concentrations (Jin et al., 2018, 2019a).



Most of these dust emission inversion systems use variational methods to estimate the optimal emissions. Since a large programming effort is required to formulate and implement the tangent linear (TL) model and its adjoint model (AM) in the traditional 4DVar, those systems often employ model-reduced or ensemble-based variational assimilation. With model reduction, a simplified tangent-linear model is used to propagate the background error covariance. Ensemble methods generate

an ensemble of perturbed emissions and propagate this ensemble to approximate the evolution of background error covariance. Both of these adjoint-free methods are able to reduce uncertainty in emissions by determining the dominant and sensitive patterns. The computation costs necessarily limit the size of the reduced tangent-linear model or the size of the ensemble to a number that is much smaller than the size of the emission parameter space. Consequently, the optimal emission that can be calculated are constrained to a subset of the original space, which is defined by the model or parameter reduction that was

applied.

A crucial element of all inversion methods is the proper specification of the spread in possible estimates, which is in this application the spread in possible emissions. Ideally, the emission uncertainties should be both physically reasonable and capable of providing sufficient variations to explain the observation-minus-simulation differences. Unfortunately, the many possible errors that could be present in dust emission parameterizations could not be described all together, and simplifications

are needed. Many studies use fairly coarse emission uncertainty, limited to optimization of a few scaling factors for emission inventories spanning a larger domain. For example, in the dust emission inversion research by Yumimoto et al. (2008), the emission background covariance is assumed to be uncorrelated in space and the uncertainty is simply defined as 500% of the prior emission flux rate. Khade et al. (2013) introduced an uncertain erodibility fraction parameter field to introduce variability in dust emissions over the Sahara desert, and reduced the uncertainty by using an Ensemble Adjustment Kalman filter (EAKF).

Di Tomaso et al. (2017) attributed the emission error to the uncertainty in the Friction Velocity Threshold (FVT), which was reduced by estimating an optimal correction factor using a Local Ensemble Transform Kalman Filter (LETKF). Limited by the ensemble size, the multiplicative value was considered spatially and temporally constant. In a previous study described in Jin et al. (2018), a spatially varying multiplicative factor was applied to compensate the errors in the FVT in the dust emission parameterization. More recently in Jin et al. (2019b), the uncertainties were described by including uncertainty in the FVT and

in the surface wind field.

An essential step before starting an inversion is to check whether the specified uncertainties are actually able to explain the differences between observations and simulations. The sensitivity of the model with respect to the uncertainties should learn if the parameters considered are really the dominant problematic parameters. Under the circumstances that the aforementioned model-reduced or ensemble-based variational data assimilation algorithms are adopted, the knowledge of the sensitivity is

particularly valuable, since it can efficiently help the model/parameter reduction by removing those insensitive problematic parameters. Based on this knowledge, the background covariance could be improved which will immediately improve the emission inversions.

An efficient way of examining sensitivities is the use of an adjoint model. This is especially useful to examine the sensitivity of a limited number of output values for changes in a large amount of input values. The first implementations of an adjoint of an

atmospheric transport model was in the early 1980's with applications in numerical weather forecasting (Dimet and Talagrand,





1986; Talagrand and Courtier, 1987). Implementations in chemical transport models (CTMs) can be found in (Elbern et al., 1997; Hakami et al., 2005; Hourdin and Talagrand, 2006; Henze et al., 2007b; An et al., 2016). The standard forward version of a CTM requires input from initial conditions and model parameters, and provides concentrations in receptor points as output. The state evolution could therefore be regarded as source-oriented. Adjoint models, however, could be regarded as receptor-

oriented, as they use a distortion in a receptor point as input, and compute from this the distortions of the input parameters that explain this. In case of many uncertain parameters, an adjoint model is very efficient in calculating model sensitivities than other methods such as the traditional finite-difference method, which requires many forward model runs with perturbed inputs (Zhai et al., 2018).

    In this study, we first review the emission inversion conducted in Jin et al. (2019b), where the Himawari-8 satellite AOD

observations was assimilated for a dust storm event in May 2017. Although significant improvements on dust simulation and forecast skills driven by the posterior emissions were reported, some large regional simulation errors remained. In particular, during three severe dust outbreaks (SD), some high dust concentrations observed at ground level were not at all or not completely resolved by the *a posteriori* simulations, although the assimilated AOD observations also indicated that a severe dust plume was present. An adjoint version of the transport model is then introduced. It will not be used to optimize emis-

sions (although that would make sense in a 4DVar context), instead it is used to trace back the potential emission source that could explain the observed high concentrations. For the three selected dust outbreaks the sensitivity towards the emissions is computed for observation sites that were not resolved correctly by the assimilation. Each of the results pointed at the Horqin desert (or Horqin sandy land) as the most likely source region for this event. Up to now, this desert was considered to be of less importance as source region (Zhang et al., 2003), and is not present as an easily erodible in the dust emission scheme

included in our dust model. To evaluate whether dust emissions from the Horqin desert could indeed explain the observed high concentrations, a new inversion is applied with a modified emission model with a higher surface erodibility over this region. The new reference model is further improved by assimilating ground based $PM_{10}$ observations, which significantly reduce the remaining differences.

    While various studies on aerosol and/or dust emission inverse modeling assume that the location of sources is known, this

study represents application of this methodology in detecting dust source areas which are still not recognized as sources with significant contribution to airborne dust cycle. Within this context, the highlights are twofold. First, this study shows how an adjoint model could be used to identify potential sources in case large observation-minus-simulation error residues are found that cannot be explained by the existing model and assumed or empirical uncertainties, and thus cannot be corrected using a data assimilation system. With the potential source region identified by the adjoint sensitivities, the background emission

uncertainty is updated. Second, although the existing emission scheme worked properly in most deserts in East Asia, e.g., Gobi and Mongolia, it highly underestimated the possible emissions from the Horqin desert. Based on our results, it is advised that emission sources in dust transport models include Horqin desert as a more active source region.

    This paper is organized as follows: Section 2 mainly discusses the numerical dust transport model, the various causes of simulation emission errors, and the difficulties in accurate emission uncertainty quantification. Section 3 reviews the emission

uncertainty construction that was used in a previous study (Jin et al., 2019b) on dust storm emission inversion for an event in



**Table 1.** Dust aerosol size distribution in LOTOS-EUROS.

| Bins | dust_ff | dust_f | dust_ccc | dust_cc | dust_c |
|---|---|---|---|---|---|
| Diameter range ($\mu$m) | 0.01 to 1 | 1 to 2.5 | 2.5 to 4 | 4 to 7 | 7 to 10 |

May 2017. Section 4 shows the locally high error residues in the assimilation found in the previous study, when three severe dust outbreaks are not well reproduced in northeast China even though the assimilated measurements indicated severe dust plumes. Section 5 presents the theory of adjoint modelling and how to detect the potential emission source for the three dust outbreaks. In Section 6, the dust model is reconstructed by assuming higher soil erodibility for emissions over the potential source regions found with the adjoint model. The emission uncertainty is also updated here. Finally, a regional emission inversion is performed again using the new input. Section 7 further discusses the added value of using adjoint sensitivities for detecting sources to resolve observation-minus-simulation errors.

## 2 Emission error analysis

### 2.1 Dust model

A regional chemical transport model, LOTOS-EUROS, is used to simulate the dust life cycles including emission, advection, diffusion, dry and wet deposition, and sedimentation (Manders et al., 2017). To simulate dust outbreaks in East Asia, the model is configured on a domain from 15°N to 50°N and 70°E to 140°E, at a resolution of 0.25°× 0.25°. Vertically, the model is configured on 8 layers with a top at 10 km, where the second layer is a mixing layer representing a well mixed boundary layer. The model is driven by meteorological data from the European Center for Medium-Ranged Weather Forecast (ECMWF), in this study operational forecasts for horizons of 3-12 hours starting from the 00:00 and 12:00 analyses, retrieved at a regular longitude/latitude grid of about 7 km resolution. The dust aerosols in the model are described by 5 aerosol bins as shown in Table.1.

The severe dust storm event studied in this paper took place over east Asia in May 2017, and has already been used as case study for emission inverse modeling in Jin et al. (2019b). The event was reported to be an extreme severe one, with dust concentrations at downwind cities reaching up to 2,000 $\mu$g/m$^3$. After crossing north China, the dust plume moved further east to the Korean peninsula and Japan (Minamoto et al., 2018), and part of the plume was eventually even transported across the Pacific Ocean (Zhang et al., 2018).

### 2.2 Emission parameterization and error analysis

The physical basis of the dust emission model adopted in LOTOS-EUROS is the parameterization scheme by Marticorena and Bergametti (1995). The dust flux $\mathcal{F}_v$ is calculated as a function of horizontal saltation $\mathcal{F}_h$, the sandblasting efficiency $\alpha$ (Shao et al., 1996), a terrain preference $\mathcal{S}$, and an erodible surface fraction $\mathcal{C}$ as:

$$\mathcal{F}_v = \mathcal{F}_h \cdot \alpha \cdot \mathcal{S} \cdot \mathcal{C} \tag{1}$$





The dust saltation rate $\mathcal{F}_h$ is proportional to the third power of the wind friction velocity $u_*$, as long as this exceeds a certain (surface depended) friction velocity threshold $u_{*t}$:

$$
\mathcal{F}_h =
\begin{cases}
& u_* \leq u_{*t} \\
\frac{p_a}{g}\, u_*^3 \left(1 + \frac{u_{*t}}{u_*}\right)\left(1 - \frac{u_{*t}^2}{u_*^2}\right) & u_* > u_{*t}
\end{cases}
\tag{2}
$$

The friction velocity $u_*$ is computed from the ECMWF 10 m wind speed assuming neutral atmospheric stability, following
a logarithmic profile. The friction velocity threshold $u_{*t}$ is derived first for an idealized dry and smooth surface, and then increased using two correction factors that described the actual situation in a grid cell: the first factor accounts for soil moisture in presence of clay, the second factor accounts for surface roughness elements. More formulas and details related to the $\mathcal{F}_h$ parameterization can be founded in Jin et al. (2018).

Of the other factors in Eq.1, the sandblasting efficiency $\alpha$ is determined by the average diameter of the soil particles in salta-
tion and the average diameter of suspended particles. The terrain preference $\mathcal{S}$ represents the probability of having accumulated sediments in a given model cell (Ginoux et al., 2001), calculated as:

$$
\mathcal{S}_i = \frac{z_{max} - z_i}{z_{max} - z_{min}}
\tag{3}
$$

where $z_i$ denotes the elevation of the given grid cell $i$, while $z_{max}$ and $z_{min}$ represent the maximum and minimum elevations in the surrounding $10° \times 10°$ area, respectively. The current configuration assumes that only area's that are identified as barren
surfaces in the landuse maps allow wind blown dust emissions, while all vegetated or water covered surfaces are considered as non-erodible. The fraction of barren surface $\mathcal{C}$ in a grid cell is taken from the Global Land Cover database (http://forobs.jrc.ec. europa.eu/products/glc2000/).

Even though the existing parameterizations were already validated with a high credibility either in wind tunnel tests or in simulations for case studies, the representation of these schemes in regional and global atmospheric models are still limited.
Many uncertainties are present, for example in the landuse (derived from Global Land Cover database) and soil data bases (derived from The PSU/NCAR mesoscale model (known as MM5)) that are used as input. These uncertainties result in differences between observations and simulations that cannot be traced back immediately to a single cause. Besides, these deterministic parameterizations are not representative for the stochastic nature of dust emissions. For example, the dust saltation only occurs when $u_t$ exceeds the minimum friction velocity that is needed to initiate a movement of soil particles. However, observations
show that within the dust particle size range the threshold friction velocity also differs widely due to stochastic inter-particle cohesion. In reality there will always be a (small) amount of free moving dust which can be resuspended even by weak wind forces (Shao and Klose, 2016).

Several emission inverse modeling studies have analyzed and estimated sources of dust aerosols on regional scales, and decreased uncertainties in the emission model by minimization of observation-minus-simulation differences. However, the
large amount of uncertainties cannot be constrained completely by the available observations. Most studies therefore coarsen the uncertainties, limiting the optimization to only a few scaling factors for the emissions field (e.g., Yumimoto et al. (2008)) or for precursor emission inputs (e.g. for the relative erodibility surface fraction by Khade et al. (2013) and friction velocity





threshold by Di Tomaso et al. (2017); Jin et al. (2018)) spanning large domains. However, a coarse and simplified emission uncertainty configuration might not be able to resolve all observation-minus-simulation differences during the inverse modeling. An example of this will be shown in Section 4, where three severe dust outbreaks are described. The emission inversion assimilating satellite aerosol optical depth (AOD) was able to produce *a posteriori* emission fields that lead to dust simulations

in agreement with dust observations at ground level, except for small region in the domain. The next section first describes the inversion system that was used.

## 3   Dust emission inversion

The dust storm event over east Asia that took place in May 2017 has been used as case study for data assimilation in (Jin et al., 2019b). In that study, an assimilation system around the same transport model (LOTOS-EUROS) was used to assimilate

AOD observations from the Advanced Himawari Imager (AHI) instrument on board of the geostationary Himawari-8 satellite (Yoshida et al., 2018). The AHI instrument provides observations with a fine temporal (10 minute) and spatial (5 kms) resolution, and a wide domain covering the East Asia. The Himawari-8 aerosol products have been widely used in the airborne aerosol data assimilation (Yumimoto et al., 2016b; Sekiyama et al., 2016; Dai et al., 2019). The assimilation system adjusted the dust emissions in the source regions to obtain the best comparison between simulated and observed AOD. Through com-

parison with independent $PM_{10}$ data, the dust concentration forecast was validated to be strongly improved at most downwind sites by the assimilation.

The uncertainty of the emission in (Jin et al., 2019b) was mainly assigned as a sum of two sources, the uncertainty in the friction velocity threshold and in the erosive wind fields. The uncertainty in the friction velocity threshold $u_{*t}$ was described by a spatially varying multiplicative factor $\beta$, defined as random variables with a mean of 1.0 and a standard deviation $\sigma$ of 10%.

The uncertainty in the friction wind velocity $u_*$ was described by the spread in a meteorological ensemble with 26 members. Note that the dust emission model computes hourly emissions per grid cell, which may vary strongly from hour to hour. In the inversion system, the temporal variation of the emission model is maintained and could be further increased by the uncertainty during the assimilation window(s) of 24 hours.

Fig. 1a shows the accumulated dust emission flux from May 02 15:00 to May 04 15:00 China Standard Time (CST). These

dust emissions are responsible for the event that is studied. Outside of this period, the dust emissions are rather weak. The figure shows that the main source regions are in the Gobi and Mongolia deserts. Fig. 1b shows the corresponding standard deviation of the accumulated emission that follows from assumed uncertainty.

Snapshots of Himawari-8 AODs are shown in Fig. 2. This type of data was assimilated with LOTOS-EUROS simulations in two 24 h windows. The posterior accumulated emission are also shown in Fig. 1c. Both the prior and posterior simulation indi-

cate that the dust was emitted from the Gobi, Mongolia and Alex deserts. Previous research (Zhang et al., 2018) and simulation from an operational dust forecast model, BSC-DREAM8b (https://ess.bsc.es/bsc-dust-daily-forecast), have identified the same emission source for this event. eThe red box in Fig. 3 indicates the location of the Horqin desert. The area is not a completely sandy desert but has some vegetation, although sparse. No (or hardly) any dust emissions are assumed to be released from here




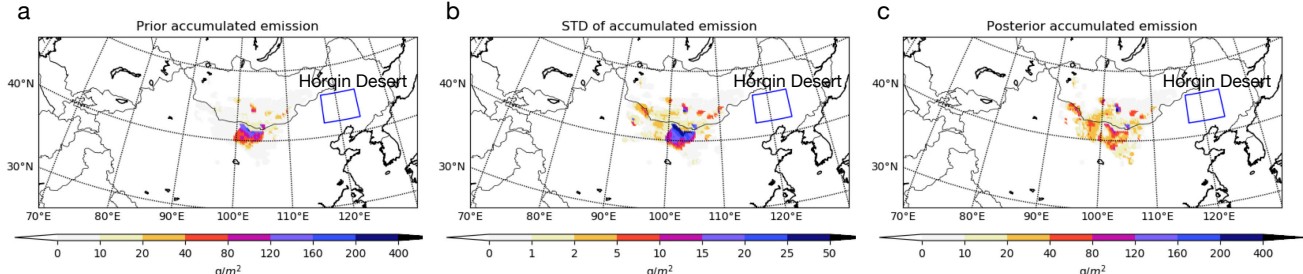

**Figure 1.** Accumulated prior dust emissions from 2017 May 02 15:00 (CST) to May 04 15:00 (a), as well as the assumed standard deviation (b), and the estimate after assimilation (c). This figure is adapted from Fig.2 in Jin et al. (2019b).

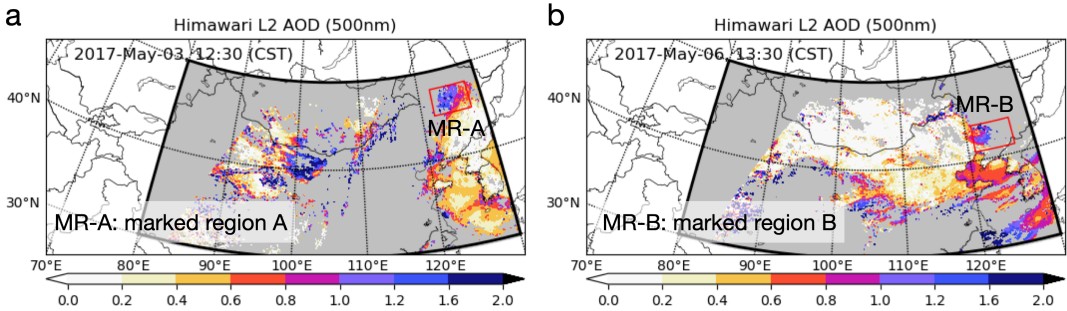

**Figure 2.** Two snapshots of Himawari-8 Level 2 AODs (500 nm) at May 03 12:30 and May 06 13:30. Note that only observations within the black framework are included, where gray values denote pixels for which no AOD was retrieved.

in the emission model, and therefore also the associated uncertainty is zero. Thus, the Horqin desert is in the model considered as completely free of dust emissions, and emissions could also not be introduced by the inversion system. However, as we shall see later on, dust emissions from this region could very well explain observed differences between observations and simulations, and therefore the inversion system should be adjusted to allow emissions from there too.

5    Dust concentration forecasts based on the *a posteriori* emissions have been validated by comparison with ground based PM$_{10}$ measurements. Snapshots of the *a posteriori* surface dust concentrations as well as PM$_{10}$ measurements are shown in Fig. 4 and Fig. 5.

## 4   Regional differences between observations and simulations

As of yet, over 1,500 field stations all over China have been established by the China Ministry of Environmental Protection
10    (MEP) to monitor atmospheric constituents including PM$_{2.5}$, PM$_{10}$, CO, O$_3$, SO$_2$ and NO$_2$. The observation network is shown in Fig. 3. Hourly averaged PM$_{10}$ observations from the network are used as independent data to evaluate the *a posteriori* dust

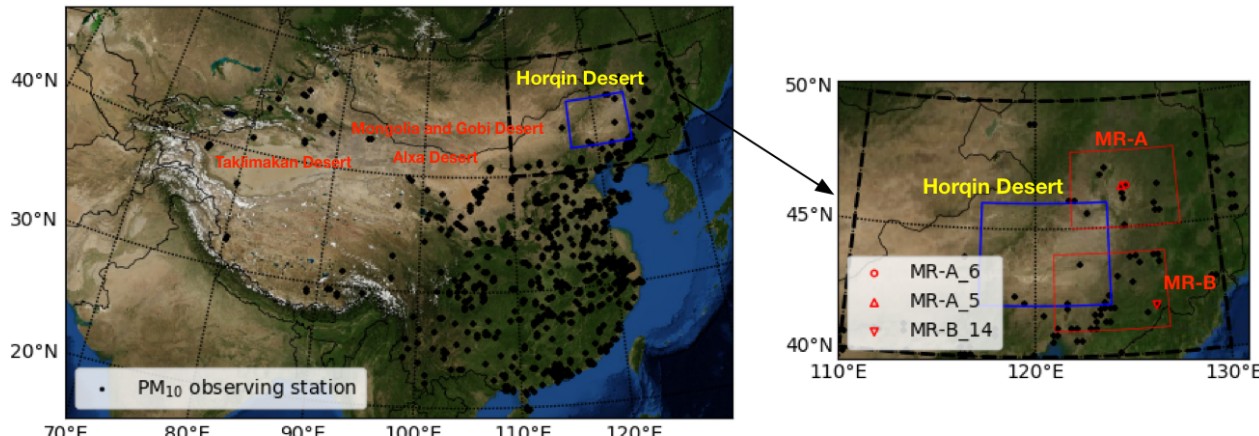

**Figure 3.** Locations of the Mongolia and Gobi, Alxa, Taklimakan, and Horqin deserts. The dots indicate locations of China MEP air quality monitoring sites. Red marked region A (MR-A) and marked region B (MR-B) are where the dust is observed but not reproduced using the transport model in this study.

simulations after assimilation of Himawari-8 AOD. Although these $PM_{10}$ measurements are actually a sum of the dust aerosols and particles released by anthropogenic activities, the values are dominated by dust during the severe events that are studied here. Therefore, all these measurements are assumed to be representative for comparison with the dust simulations. In case of less severe dust storms, observational bias corrections (Dee and Uppala, 2009; Jin et al., 2018, 2019a) would be required to
remove the *non-dust* part from the observations to allow comparison with a 'dust-only' model.

Although for most locations the *a posteriori* dust simulations showed good agreement with the $PM_{10}$ observations, some large mismatches remained, especially in the northeast part of China. Specifically, extremely high values of surface dust concentration over three severe dust events were reported by the ground-based monitoring system in this region, but neither could be reproduced to full extent by the simulations. This is illustrated in Fig. 4 for the first severe dust plume from 2017 May 03
08:00(CST) to 20:00, which we will refer to as "SD1", and in Fig. 5 for the second dust outbreak from May 04 02:00 to 14:00, which is referred as "SD2". Similar figures for the third events ("SD3") are available as supplementary material.

The top row in Fig. 4 shows $PM_{10}$ observations at three different moments during the SD1 event. Obviously a dust plume that crosses the red marked region A (MR-A), with maximum $PM_{10}$ observations rising rapidly from 200 $\mu g/m^3$ at 08:00 to more than 2,000 $\mu g/m^3$ at 20:00. The second and third row show the *a priori* and *a posteriori* LOTOS-EUROS simulations on
the surface dust concentration for the same hours. Unfortunately, the simulations in the MR-A region were completely free of dust in both simulations. Note that the simulated prior and posterior AODs, which are not shown here, generally have a similar profile as the surface dust concentration shown in Fig. 4(b) and (c).



The Himawari-8 AOD maps also indicated the existence of a severe dust plume over MR-A, as can be seen in the snapshot of AOD at May 3 12:30 in Fig. 2(a). Most of the AOD values over MR-A exceed 1.2. Our first 24 h cycle of emission inversion was performed by assimilating these high-valued AOD. The simulations driven by the posterior emission fields, shown in Fig. 4(c.1) ~(c.3), did however not lead to a dust load over MR-A during this period. The difference between the posterior

simulations and observations indicate that the current emission model and associated uncertainties cannot explain the dust plume in MR-A. In other words, the dust plume moved over MR-A was not due to emissions from the Gobi and Mongolia deserts we predefined in the background emission, but must originate from somewhere else.

The three snapshots of $PM_{10}$ observations in Fig. 5(a) indicate the second severe dust plume (SD2) over the same region MR-A. In this case, both *a prior* and *a posterior* LOTOS-EUROS model simulations include a dust plume over MR-A (see

Fig. 5(b) and Fig. 5(c) ), which could be traced back to emissions from Gobi, Mongolia, and Alex deserts. The maximum of the modeled surface dust concentration over MR-A on May 4 is around 500 $\mu g/m^3$. However, the maximum $PM_{10}$ measurement value exceeds 2,000 $\mu g/m^3$. It is true that these observation-minus-simulation might be caused by the emission underestimation over the Mongolia and Gobi deserts. Yet those emissions also contributed the dust plume observed in Central China. In this case, those dust flux rates are actually constrained at a modest level by those observations. Besides, the dust plume did not fully

cover the observed dust-affected regions. Thus, the dust level is considered to be partially due to the predefined emissions, but also due to emissions from another region. For this event, Himawari-8 measurements are not successfully retrieved due to cloud scenes over MR-A, thus AOD snapshots are not available.

The underestimation of dust concentrations over MR-A during the SD1 and SD2 events was also found in other simulation systems, for example as published by the SDS-WAS service (https://ess.bsc.es/bsc-dust-daily-forecast). As example, results

for SD1 and SD2 from the forecast system BSC-DREAM8b (Basart and Carlos, 2012; Mona et al., 2014) are shown in the last row of Fig. 4 and Fig. 5, respectively. These suggests that these emission models are also prone to underestimate the emission rate over Horqin desert.

Similar conclusion was drawn for the third dust outbreak ("SD3"), for which simulation and $PM_{10}$ measurements are available in the supplementary material. For SD3, it was found that severe dust plume was recorded over the marked region (MR-B)

in the northeast China. However, neither the *a prior* nor the *a posterior* simulations of the BSC-DREAM8b simulation reproduce any dust over MR-B, although the assimilated Himawari-8 AOD values did indicated the existence of a dust plume over this region, as shown in in Fig. 2(b).

To further illustrate the three severe dust outbreaks in the Northeast China on May 03 and 04, the time series of the $PM_{10}$ observations averaged over all monitoring stations inside the marked regions MR-A are shown in Fig. 6(a). The average $PM_{10}$

levels are around 100 ~200 $\mu g/m^3$ when there is no dust (earlier than May 02 12:00). The peak of SD1 arrives in marked region MR-A around May 03 08:00, and has left the region at May 04 00:00; the averaged $PM_{10}$ concentrations have reached a value up to 1,000 $\mu g/m^3$. The most severe dust plume occurs during SD2 at May 04, with average $PM_{10}$ measurements inside MR-A up to 1,500 $\mu g/m^3$.

**Figure 4.** PM$_{10}$ observations and surface dust concentrations simulated for the 2nd severe dust event (SD2) for May 03 08:00 (CST) (a.1~d.1), 14:00 (a.2~d.2), and 20:00 (a.3~d.3). Top row PM$_{10}$ observations (a.1~a.3), second row prior simulations (b.1~b.3), third row posterior simulations (c.1~c.3), and bottom row BSC-DREAM8b simulations (d.1~d.3). MR-A: marked region A.



**Figure 5.** PM$_{10}$ observations and surface dust concentrations simulated for the 2nd severe dust event (SD2) for May 04 02:00 (CST) (a.1~d.1), 08:00 (a.2~d.2), and right column 14:00 (a.3~d.3). Top row PM$_{10}$ observations (a.1~a.3), second row pror simulations (b.1~b.3), third row posterior simulations (c.1~c.3), and bottom row BSC-DREAM8b simulations (d.1~d.3). MR-A: marked region A.



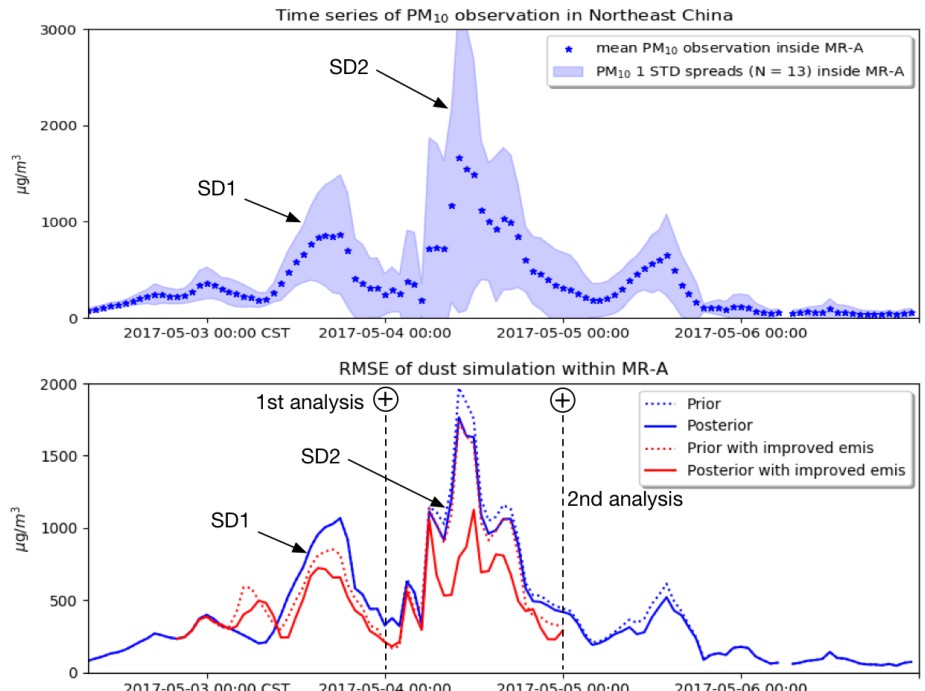

**Figure 6.** a: Hourly $PM_{10}$ observations averaged over MR-A (13 sites), with shaded area the one standard deviation range; b: root mean square error of prior and posterior simulation, reconstructed prior and posterior within MR-A. The location of the stations is indicated in Fig. 3.

## 5  Determining emissions sources using an adjoint model

The adjoint approach provides an efficient tool for calculating the sensitivity of a simulation model with respect to its input parameters. In this study, an adjoint model is used to identify potential source regions for dust that could explain the mismatch between simulations and observations in the north east of China.

### 5.1  Adjoint theory

The following notation will be used for the discrete time step of our simulation model:

$$\boldsymbol{x}^k = \mathcal{M}^{k-1}(\boldsymbol{x}^{k-1}, \boldsymbol{f}^{k-1}) \tag{4}$$

In here, $\boldsymbol{x}^k$ denotes the state vector at time $k$ that consists of 3D fields of dust aerosol concentrations for each of the 5 dust size bins in the model, input vector $\boldsymbol{f}^{k-1}$ consists of emission fields for the 5 size bins, and $\mathcal{M}^k$ denote the model operator that simulates $\boldsymbol{x}^k$ given the state and input at time $k-1$. For a pure dust transport simulation, the model is linear with respect to





both $\boldsymbol{x}$ and $\boldsymbol{f}$, and could therefore be written using matrix operators:

$$\boldsymbol{x}^k = \mathbf{M}^{k-1}\,\boldsymbol{x}^{k-1} + \mathbf{E}^{k-1}\,\boldsymbol{f}^{k-1} \tag{5}$$

The operator $\mathbf{M}^k$ represents the transport part of the model, while $\mathbf{E}^k$ represents the emission part. Repeated application of Eq. (5) provides the evolution of the state from time $k-K$ to time $k$:

$$
\begin{aligned}
\quad \boldsymbol{x}^k &= \mathbf{M}^{k-1}\left(\mathbf{M}^{k-2}\,\boldsymbol{x}^{k-2} + \mathbf{E}^{k-2}\,\boldsymbol{f}^{k-2}\right) + \mathbf{E}^{k-1}\,\boldsymbol{f}^{k-1} \tag{6}\\
&= \mathbf{M}^{k-1}\cdot\mathbf{M}^{k-2}\,\boldsymbol{x}^{k-2} + \mathbf{M}^{k-1}\cdot\mathbf{E}^{k-2}\,\boldsymbol{f}^{k-2} + \mathbf{E}^{k-1}\,\boldsymbol{f}^{k-1} \tag{7}\\
&= \mathbf{M}^{k-1}\cdot\mathbf{M}^{k-2}\cdot\ldots\cdot\mathbf{M}^{k-(K-1)}\cdot\mathbf{M}^{k-K}\,\boldsymbol{x}^{k-K} + \tag{8}\\
&\quad\ \mathbf{M}^{k-1}\cdot\mathbf{M}^{k-2}\cdot\ldots\cdot\mathbf{M}^{k-(K-1)}\cdot\mathbf{E}^{k-K}\,\boldsymbol{f}^{k-K} + \tag{9}\\
&\quad\ \ldots + \mathbf{M}^{k-1}\cdot\mathbf{E}^{k-2}\,\boldsymbol{f}^{k-2} + \mathbf{E}^{k-1}\,\boldsymbol{f}^{k-1} \tag{10}
\end{aligned}
$$

We define a model *response function* as a scalar function of the state:

$$\mathcal{J}(\boldsymbol{x}^k) \in \mathcal{R} \tag{11}$$

The response could for example be defined as the simulation at a single location (an observation site), or an average over multiple grid cells. The gradient of this response function at time $k$ with respect to the input vector $\boldsymbol{f}^{k-K}$ follows from the application of the chain rule, and using that Eq. (9) is the only term in the expansion of $\boldsymbol{x}^k$ that depends on $\boldsymbol{f}^{k-K}$:

$$
\begin{aligned}
\quad \nabla_{\boldsymbol{f}^{k-K}}\mathcal{J}(\boldsymbol{x}^k) &= \nabla_{\boldsymbol{f}^{k-K}}\left(\boldsymbol{x}^k\right)^T\cdot\nabla_{\boldsymbol{x}^k}\mathcal{J}(\boldsymbol{x}^k) \tag{12}\\
&= (\mathbf{E}^{k-K})^T\cdot(\mathbf{M}^{k-K})^T\cdot\ldots\cdot(\mathbf{M}^{k-2})^T\cdot(\mathbf{M}^{k-1})^T\cdot\nabla_{\boldsymbol{x}^k}\mathcal{J}(\boldsymbol{x}^k) \tag{13}
\end{aligned}
$$

The transpose $(\mathbf{M}^k)^T$ of the linear model operator $\mathbf{M}^k$ is referred as the *adjoint model*. To compute the above gradient $\nabla\mathcal{J}$, the adjoint model is applied in a reverse time sequence $k-1,\, k-2,\, \ldots,\, k-K$. The first adjoint operation in this sequence is applied on the *adjoint forcing*:

$$\nabla_{\boldsymbol{x}^k}\mathcal{J}(\boldsymbol{x}^k) \tag{14}$$

     An adjoint model is a powerful tool to compute the model response with respect to various input parameters. A useful application is found in 4D variational data assimilation, where it is used to derive the gradient of a cost function for the difference between observations and simulations. In the context of air quality, this approach has been used to constrain initial conditions, emissions, and other uncertain model parameters such as uptake (Elbern et al., 2000; Henze et al., 2009).

For this study, an adjoint implementation of the LOTOS-EUROS model will be used to identify potential emission source regions. The adjoint model is created from the same source code, but using an internal flag it applies adjoint (transpose) versions of the transport and emission operators. Using a negative time step it is able to run backwards in time, as is required to compute the gradients as in Eq.13. The assimilation system that is used in this study remains the reduced-tangent-linearization 4DVar that was developed in earlier studies Jin et al. (2018, 2019a, b), which does not use the adjoint implementation. Although it 30  would be possible to use the adjoint for the assimilation too, it was chosen to keep the assimilation system the same in order to compare results before and after the introduction of new emission sources.





## 5.2 Testing the implementation of the adjoint model

Before using the adjoint model to identify potential emission sources, the implementation is first illustrated and tested by looking at a single site.

A suitable test to validate whether the adjoint model computes the correct sensitivity of the model towards changes in the
input, is to compare its evaluation with a finite-difference method (Henze et al., 2007a; Guerrette and Henze, 2015). That is, the sensitivity of a model response $\mathcal{J}(\boldsymbol{x}^k)$ to the previous emission field $\boldsymbol{f}^{k-K}$ is computed either using either the adjoint, or by perturbing the emission field. For this test, we define the response function to be the dust concentration in the grid cell where during the most severe dust plume (SD1) the highest $PM_{10}$ concentration was observed within marked region MR-A, referred to as "MR-A_6", the location of which could be found in Fig.3. The response function becomes:

$$\mathcal{J}(\boldsymbol{x}^k) = \mathbf{H}\boldsymbol{x}^k \qquad (15)$$

where the matrix operator $\mathbf{H}$ is actually a row vector with zeros except for the elements that represent the 5 dust size bins in the selected grid cell:

$$\mathbf{H} = [0,\ldots,0,1,1,1,1,1,0,\ldots,0] \qquad (16)$$

The adjoint forcing becomes:

$$\nabla_{\boldsymbol{x}^k}\mathcal{J}(\boldsymbol{x}^k) = \mathbf{H}^T \qquad (17)$$

Time $t_k$ is set to 19:00 on May 3 2017 when the dust concentration in MR-A_6 peaked.

Following Eq. (13), the sensitivities of this dust concentration towards dust emissions at time $t_{k-K}$ is:

$$\nabla_{\boldsymbol{f}^{k-K}}\mathcal{J}(\boldsymbol{x}^k) = (\mathbf{E}^{k-K})^T \cdot (\mathbf{M}^{k-K})^T \cdot \ldots \cdot (\mathbf{M}^{k-2})^T \cdot (\mathbf{M}^{k-1})^T \cdot \mathbf{H}^T \qquad (18)$$

A snapshot of the adjoint emission sensitivities at May 03, 13:00 CST, is shown in Fig. 7(a) for one of the 5 dust size bins in the
model. According to these values, the dust concentration in MR-A_6 simulated for 6 hours later is most sensitive to emissions that are roughly in the rectangular box. Note that in this example the response function $\mathcal{J}$ has units of concentrations, which gives $\nabla_f\mathcal{J}$ the units of concentrations ($\mu$g/m$^3$) over emissions ($\mu$g/m$^2$/s), equivalent to s/m.

The same sensitivity could also be calculated using a finite-difference method. For this, 16 locations are chosen within the box shown in Fig. 7(a). The locations are marked with dots, and put at locations where the adjoint sensitivities are non-zero.
Then 16 model runs are performed over [13:00,19:00], where each run is similar to a standard simulation, but using emissions that are only non-zero at [13:00,14:00] at just one of the 16 marked locations. The magnitude of these emissions is simply set to 1 $\mu$g/m$^2$/s for each bin. The result of each simulation is the simulated concentration in $\mu$g/m$^3$ in the MR-A_6 location at 19:00. The ratio between simulated concentration and emission has units $s/m$ and is a measure for the sensitivity of the simulation in MR-A_6 at 19:00 towards an emission at one of the marked locations at 13:00.
The scatter plot in Fig. 7(b) compares the 16 computed sensitivities (for each of the 5 size bins) versus the sensitivities computed with the adjoint model. The results show that the adjoint-computed sensitivities are in good agreement with the





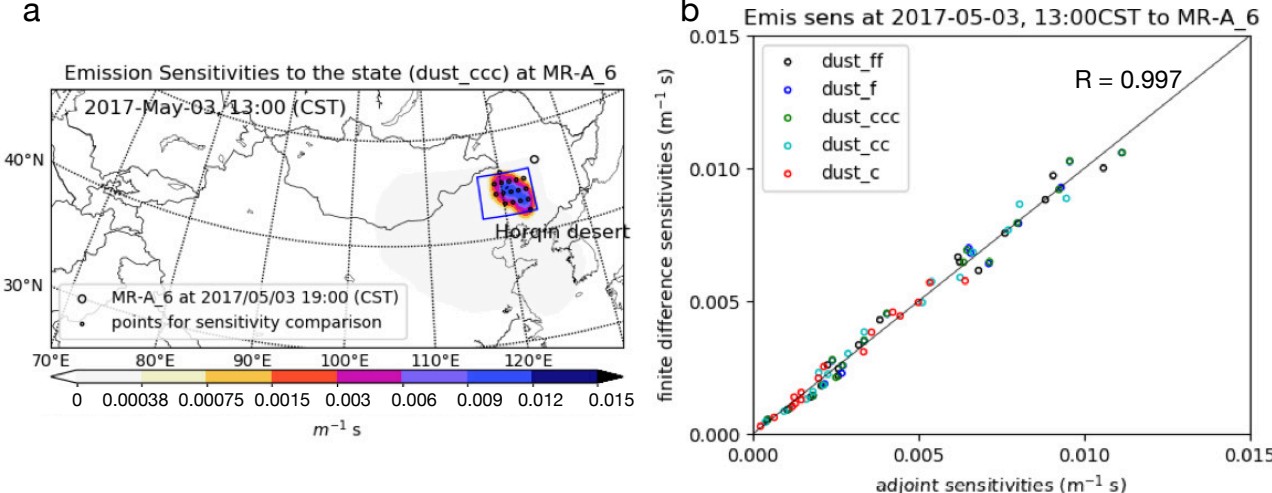

**Figure 7.** Illustration of sensitivities of dust concentrations at 19:00 in location MR-A_6 towards emissions from selected points at 13:00; (a): map of emission sensitivities computed by adjoint model; (b): comparison between sensitivities computed by adjoint method and finite differences.

finite difference sensitivities, which results in a relative high *Pearson* correlation coefficient of $R = 0.997$. The comparison suggests that the adjoint model has been implemented correctly. The differences that remain might be due to rounding errors at points where the sensitivity is low, and model processes other than transport and emission which are not included in the adjoint. Both the finite difference and adjoint method seem able to derive emission sensitivities. An advantage of the adjoint method however is that it computes sensitivities with one single simulation, while the fine difference method requires many more (16 in this example).

### 5.3 Identification of emission sources

During the investigated severe dust outbreaks (SD1 and SD2), the emission inversion was not able to provide *a posteriori* simulations that correctly represented the high dust concentrations observed in sites in the north east of China. To identify whether this could be due to missing dust sources, the adjoint model is used to identify potential source regions.

Similar as for the illustrative example in Section 5.2, the sensitivity of a response function towards changes in emissions is computed using the adjoint model, for each of the 3 dust outbreaks. The adjoint forcing $\mathbf{H}^T$ in Eq. (18) are chosen as the observed state variables in MR-A_6 on May 03 19:00 for SD1, in MR-A_5 on May 04 10:00 for SD2, in MR-B_14 on May 06 18:00 for SD3, respectively. The location of MR-A_6, MR-A_5 and MR-B_14 can be found in Fig. 3(b). These three sites (and also the surrounding stations) reported the highest PM$_{10}$ levels during the three dust outbreaks. For each case, the adjoint forcing $\mathbf{H}^T$ is filled with values of 10 $\mu$g/m$^3$ for each bin in the cell with the observation site. Time series of emission





sensitivity fields are shown in Fig. 8 for the severe dust outbreaks SD1 and SD2, while the sensitivities for SD3 are reported in the supplementary material.

Figures 8(a.1)~(a.6) show the potential source regions for the high $PM_{10}$ values observed in MR-A_6 on May 03 11:00. The blue marked box encloses the Horqin desert, which is a potential source region for dust emitted 10 hours before the observation

time. If the dust was emitted earlier, it seems to originate from regions further south. However, these are densely populated regions covered with vegetation, and therefore not a likely to be a source of dust. The sensitivity maps show that for this time period the MR-A_6 location is not sensitive for dust emitted from the Gobi and Mongolia deserts, which are in the current emission model the main source regions. This explains also why the assimilation system, that was based on adjusting emissions from these deserts, was not able to resolve the high dust levels within marked region MR-A during SD1.

As shown in Fig. 8(b.1)~(b.6), a potential source region for dust observed in marked region MR-A during SD2 is again the Horqin desert, in case the emission took place 12 hours before observation. For emissions longer ago, the Gobi and Mongolia deserts could be source regions too. According to the reference and posterior dust simulation in Fig. 5, the dust plume that originated from the Gobi desert was in fact carried to MR-A on May 04, after 20 to 30 hours of long range transport. However, the simulated dust concentrations in this plume are much lower than the observed $PM_{10}$ concentrations. The best explanation

is that the dust plume was first released from the Gobi desert, and a part of it was carried to northeast China by the prevailing winds. When it crossed over the Horqin desert, huge amounts of new dust particles were lifted too, and the mixed plume reached marked region MR-A on May 4. An observational study mainly based on Himawari-8 RGB imagery carried by Minamoto et al. (2018) also indicated that the dust particles in SD2 were not only from the Gobi desert, but might also originate from the Horqin desert, which was up to now not recognized as a potential source by most dust emission models.

Similar conclusions were drawn for the severe dust event ("SD3"), for which figures of backward emission sensitivities are available as supplementary material. For SD3, it was noticed that dust emissions from the Horqin desert between May 06 09:00 to 15:00 could explain the high dust loads observed. Earlier emissions are traced northwards from regions in Siberia that are still not identified as active dust sources.

The simulation of the emission source sensitivities over the three independent dust events all indicated that the Horqin desert

is likely to be the the main source region for SD1 and SD3, and also at least partly a source region for SD2. Therefore, the existing emission scheme needs to be adjusted to allow dust emission from the Horqin desert, especially when dust is observed in north east China.

## 6 Emission inversion with improved emission uncertainty

Parameterization of source areas, which requires knowledge on soil properties and vegetation cover, parameterization of surface

roughness, dust emission and transport processes, are some possible reasons why the current simulation model is not always able to simulate the actual dust emissions. From the study with the adjoint model it was shown that a lack of emissions from the Horqin deserts is likely to be one of these reasons. To allow dust emissions from this region too, the following changes were applied to the model the emissions and their uncertainties:

**Figure 8.** Backward time series of emission sensitivity of the dust simulation at MR-A_6 2017 May 03, 19:00 CST: emission sensitivity distribution at 2017 May 03, 18:00 (a.1), 15:00 (a.2), 12:00 (a.3), 09:00 (a.4), 06:00 (a.5), 03:00 (a.6); and of the dust simulation at MR-A_5 2017 May 04 10:00: emission sensitivity distribution at 2017 May 04, 09:00 (b.1), 05:00 (b.2), 01:00 (b.3), May 03, 21:00 (b.4), 17:00 (b.5), 13:00 (b.6).





- In the landuse data base, most parts of the Horqin desert are described as '*sparse vegetated*'. For this region, the properties of sparse vegetated surfaces are set similar as '*bare areas*', which leads to a higher erodibility parameter $\mathcal{C}_i$ in Eq.1.

- The terrain preference correction is disabled, leading to $\mathcal{S}_i = 1$ in Eq.3.

- A tuning factor 0.7 is used to obtain a lower new friction velocity threshold in Eq.2.

- The uncertainties in the new emission field is described similar as in Jin et al. (2018, 2019b) by correction factors applied to the new friction velocity threshold. The correction factors are spatially varying and have a mean 1 and a standard deviation 10%.

These changes are highly empirical, and chosen just to have better dust simulations for May 2017. However, these might not be sufficient to correctly describe the emissions from the Horqin dessert during other events. Application in other simulations
therefore requires careful inspection by the user.

The assimilation of Himawari-8 AODs described in Jin et al. (2019b) has been repeated using the new emission and uncertainty model. The experiment is set from the May 03 to May 05 with two 24-h assimilation cycles, which covers the two dust outbreaks, SD1 and SD2, respectively. As seen in Fig. 6(b), the two analyses are performed at May 04 00:00 and May 05 00:00, respectively. Each of them calculates the most likely emission fields in the past 24 hours that fits both the prior informa-
tion and available measurements. Himawari-8 AOD valuess are assimilated in the first cycle, of which the measurement error configurations are similar as in Jin et al. (2019b). However, almost no AOD values are retrieved in the second window over the MR-A region, hence the ground $PM_{10}$ observation are assimilated instead, of which the representation errors are set similar to those in Jin et al. (2019a).

The model domain is still configured on the whole East Asia from 15°N to 50°N and 70°E to 140°E shown in Fig. 3. The
computation complexity on our reduce-tangent-linearization 4DVar is generally proportional to the size of uncertain emission fields. To save the computation costs, the aforementioned new emission and uncertainty are only applied to dust emission over the Horqin deserts. While over the rest of the domain, the deterministic emission scheme described in Jin et al. (2019b) is used.

The accumulated dust emissions before and after assimilation are shown in Fig. 9. After assimilation (panel (b)), a much stronger total emission is estimated than what is computed by the updated *a priori* model (panel (a)). In comparison, the 'old'
parameterization scheme indicates that there is no dust emission at all as shown in Fig. 1. Snapshots of the dust simulations on SD1 and SD2 driven by these emissions are shown in Fig. 10 and Fig. 11 for three different times (columns), respectively; in each figure, the top row shows simulations using the reference emissions, and the bottom row using the assimilation result.

These maps could be compared to the observations and simulations using the original emission model as shown in Fig. 4 and Fig. 5. Driven by a more easily erodible emission scheme, the *a priori* simulation (see in Fig. 10(a)) generated a dust band
which originated from the Horqin desert and then carried towards the northeast crossing the MR-A. The dust simulation in Fig. 10(b) are obtained by assimilating the Himawari-8 AOD values on May 03. This *posterior* is in better agreement with the real dust load according to the $PM_{10}$ observations.

During SD2, parts of the dust concentrations in the MR-A originate from a dust plume that was lifted from the Gobi and Mongolia desert. This initial plume is the result of a LOTOS-EUROS simulation driven by the prior emission scheme.

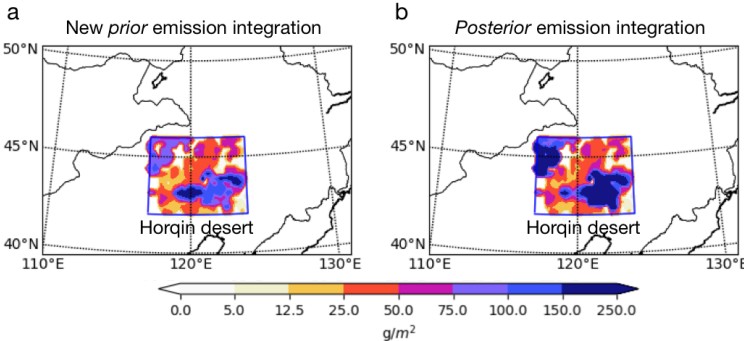

**Figure 9.** Accumulated dust emissions over the Horqin Desert from May 03, 20:00CST to May 05 07:00 (SD2): (a) prior emissions; (b) posterior emissions. The 'old' *priori* and *posterior* accumulated emission map can be seen in Fig. 1.

Meanwhile, extra particles are also mobilized from the Horqin deserts and transported northwards. The new emission model increases the dust load, however, the simulation without assimilation still under estimates the $PM_{10}$ concentrations shown in Fig. 5(a.1)~(a.3). Using the posterior emission field, the dust simulations are enhanced further, and are in much better agreement with the observations.

To quantify the improvements through the assimilation, the root mean square error (RMSE) between the observed $PM_{10}$ concentrations and the *a priori* and posterior dust simulations has been computed for each hour during the two dust outbreaks SD1 and SD2. These RMSE values are added to Fig. 6(b), which already showed similar time series for simulations using the original emission model. Using the 'new' emission model, the *a priori* RMSE values are slightly improved compared to the older simulations. Although extra emissions from the Horqin dessert are now included, the default amount is still not strong

enough to simulate the observed dust peak, especially during SD2. The largest improvement is made when assimilation is used to further enhance the emissions; the maximum RMSE values during SD1 are reduced from 1,100 to 600 $\mu$g/m$^3$, and during SD2 they are reduced from 2,000 to 1,000 $\mu$g/m$^3$. In the original assimilation configuration this could not be achieved since the emission uncertainty model did not allow any additional emissions from the Horqin desert at all.

## 7   Summary and conclusion

In this study, we illustrate the importance of using a correct background error covariance in emission inversion. An adjoint based sensitivity method is used to identify new error sources that should be included when constructing emission uncertainties. The methodology is applied to dust outbreaks over East Asia in May 2017.

First, the dust storm emission inversion in Jin et al. (2019b) was reviewed. Although in there improvements on dust simulations and forecasts have been achieved through assimilating of Himawari-8 satellite AOD, large errors still remained unresolved

at some locations. Specifically, three severe dust outbreaks in the northeast China were investigated, which are neither repro-





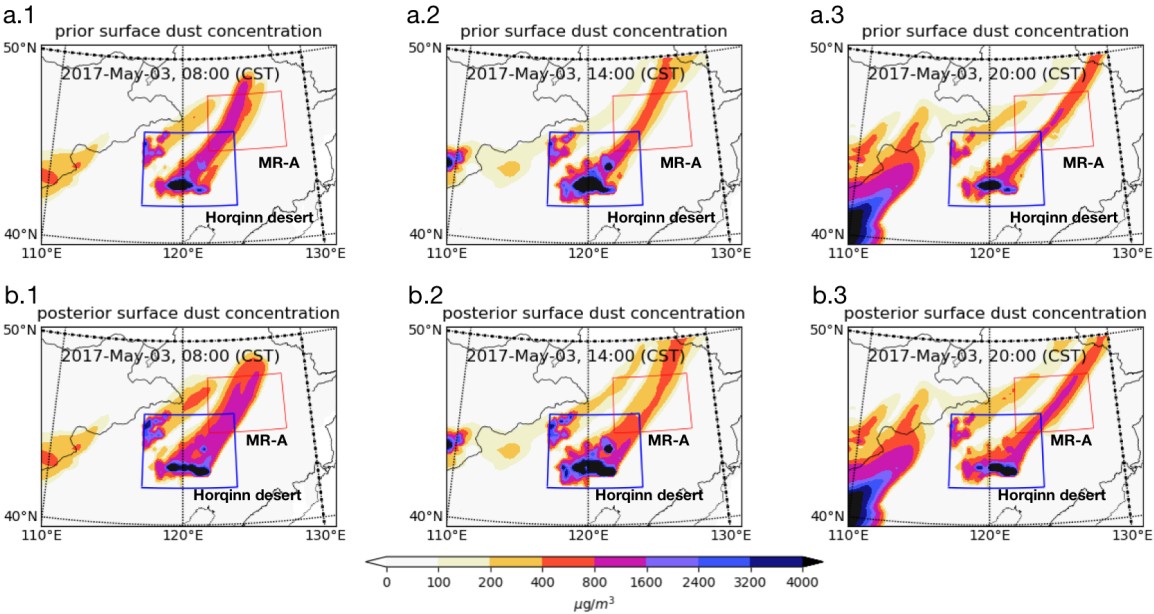

**Figure 10.** Simulation of SD1 using the new emission fields: (a) *a priori* and (b) *posterior* (by assimilating the Himawari-8 AODs) at May 03 08:00 (a.1)~(b.1); 14:00 (a.2~b.2); 20:00 (a.3 ~b.3). MR-A: marked region A.

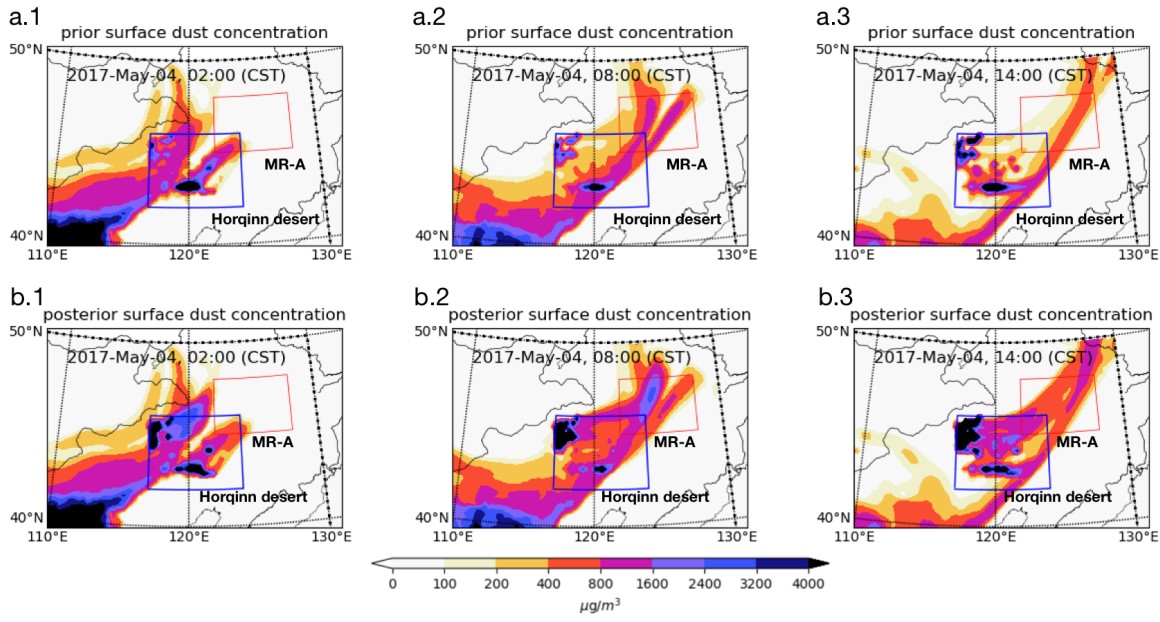

**Figure 11.** Simulation of SD2 using the new emission fields: (a) prior and (b) posterior (by assimilating the ground-based PM$_{10}$ observations) at May 04 02:00 (a.1)~(b.1); 08:00 (a.2~b.2); 14:00 (a.3 ~b.3). MR-A: marked region A.





duced by the *a priori* nor by the *posterior* simulation despite the assimilated measurements do indicate the existence of severe dust plumes.

To trace back the potential emission sources, an adjoint model has been introduced, which is efficiently calculating the sensitivities of model responses with respect to a large number of input parameters. Evaluation showed that the adjoint sensitivities

are in a good agreement with the values obtained using a finite difference method. The adjoint model was then used to trace the sensitivity of three independend dust events to emissions from upwind regions. All the experiments indicate that the Horqin desert is the most likely source regions, which is modeled as non-source in our existing emission parameterizations.

The emission scheme and the corresponding uncertainties over the Horqin desert are then reconstructed by assigning higher erodibility. The agreement with observations is only slightly improved when using a standard model simulation. However, more

significant improvements are made when a new assimilation is carried out that is able to further enhance the new emissions. The maximum RMSE between dust simulation and $PM_{10}$ observations are reduced from 2,000 to 1.100 $\mu g/m^3$. In future, the residues could be further reduced using a better reference emission as well as an improved uncertainty description for the Horqin desert. Note that also the presence of non-dust particles in the $PM_{10}$ observations limit the assimilation accuracy; removal of the non-dust part as in (Jin et al., 2019a) should become part of the standard procedure.

Although existing emission scheme work properly for most deserts in East Asia, e.g., Gobi and Mongolia, they seem to highly underestimate the Horqin desert as source region. Based on our results, it is advised that dust sources in dust transport models include Horqin desert as an sparsely vegetated active source region.

Our study clearly shows the importance of using a correct background error covariance in resolving observation-minus-simulation errors in emission inversions. The proposed adjoint method could also be performed to identify the sensitivity

towards emission sources and guide the construction of emission uncertainties. This does not only hold for applications focusing on dust, but also for other atmospheric inverse modeling applications, e.g., black carbon, haze, or gases in case that their source locations are not fully known yet.

**Data availability**

The datasets including measurements and model simulations can be accessed from websites listed in the references or by

contacting the corresponding author.

**Acknowledgments**

The real-time $PM_{10}$ data are from the network established by the China Ministry of Environmental Protection and accessible to the public at http://106.37.208.233:20035/. One can also access the historical profile by visiting http://www.aqistudy.cn/. The research product of aerosol properties (produced from Himawari-8) that was used in this paper was derived by the algo-

rithm developed by Japan Aerospace Exploration Agency (JAXA) and National Institute of Environmental Studies (NIES) and available at https://www.eorc.jaxa.jp/ptree/.





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
