# Peer review of "Source backtracking for dust storm emission inversion using adjoint method: case study of northeast China"

_Atmospheric Chemistry and Physics, 2020_

## Referee Comment (RC1) · Anonymous Referee #4 · 3 Sep 2020

The paper presents the application of an adjoint model in the context of dust data assimilation. The paper is motivated by some of the limitations shown in a previous paper by the authors (Jin et al. 2019b), where the assimilation system was not able to correct a large underestimation of dust in north east China. The adjoint model is used here to detect what the authors conclude to be the most likely source region that would explain the unresolved dust loads (Horqin desert). After adding the source in the model by increasing soil erodibility in that region, the emission inversion is performed again and it is concluded that the new results are much closer to observations. The authors also recommend to include the Horqin desert as a more active source region in models.

The study is interesting, in particular the use of the adjoint as a tool that could be used detect potentially omitted dust sources in models. However, I have some major comments (both on content and structure) that would need to be addressed before the paper can be considered for publication in ACP.

1) I find that the identification of the Horqin desert as the source that would explain the unresolved dust loads is not robust enough and would need further clarification or at least nuancing. There are few issues that make this identification uncertain:

a.    The study uses Himawari-8 AOD. How can it be concluded that the AOD signals in regions MR-A and MR-B are due mostly to dust?    The authors should provide evidence of that.    Forecasts from the US Navy on May 3, 2017 (https://www.nrlmry.navy.mil/aerosol/globaer/ops_01/mongolia/201705/2017050312_globaer_ops_mongolia.gif) show a strong influence of sulfate optical depth in addition to dust. This effect could be strongly biasing the results as it would be overestimating the dust optical depth.

b. Figure 8 shows the backward time series of emission sensitivity. The conclusion that the Horqin desert is the most likely source is based on an a priori assumption: (Page 17, line 5; "If the dust was emitted earlier, it seems to originate from regions further south. However, these are densely populated regions covered with vegetation, and therefore not a likely to be a source of dust.")  There are two problems with this statement: 1) The areas highlighted in panels a.5 and a.6 overlap with potential dust emission areas highlighted in the in-depth study and dust source inventory of Ginoux et al. (2012) (see their Figure 10, region # 6, North China Plains). 2) If the air masses are coming from the southern populated areas they may come with anthropogenic aerosols (this is consistent with the trajectory of the high sulfate optical depth from the Navy forecasts highlighted above) which goes back to my previous comment.

c. On a side note the Ginoux et al. (2012) inventory already highlights the Horqin desert as a potential dust source, and it is not the only study. Please note that the absence of emissions in your model there may be model-specific. Many models do not

assume zero emissions from sparsely vegetated areas and use LAI or fractional cover to scale emissions in those areas so they are not zero.

2) The comparison of the final results with those from the previous study (Jin et al. 2019b) after including the new source in the model is not consistent. At least that is what I understood from reading the paper. In Jin et al. (2019b) AOD is assimilated from theÂăHimawari-8 satellite AOD and PM10 is used as an independent evaluation dataset. In this paper, PM10 is also used in the assimilation (page 19, line 15: "Himawari-8 AOD values are assimilated in the first cycle, of which the measurement error configurations are similar as in Jin et al. (2019b). However, almost no AOD values are retrieved in the second window over the MR-A region, hence the ground PM10 observation are assimilated instead, of which the representation errors are set similar to those in Jin et al. (2019a)."). Therefore, the comparison between both experiments is inconsistent and the improvements (or at least a large fraction of them) stated in the text and the abstract haven't been shown to be due (at least solely) to the inclusion of the new dust source, and they could just be due to the assimilation of the PM10 (which is presumably used as the evaluation dataset as well). Experiments should be compared under the same assimilation conditions both without and with PM10.

3) Structure of the paper: I strongly recommend restructuring the paper to follow the classic introduction, data and methods, results, discussion, conclusion format. The paper combines methods, data with results and discussion thought-out the paper and this makes the reading difficult.

Additional comments or minor issues:

- Equation 3 in the original publication of Ginoux et al. (2001) is the same expression but to the power of 5.

- Page 7, line 32: there is a typo ("eThe red box")

- Again, I recommend restructuring the paper and include all the methods after the

introduction. Also in my opinion, section 5.2 interrupts the flow of the manuscript. It would be enough to say thatÂăboth the finite difference and adjoint method seem able to derive emission sensitivities and refer to an include this section in an appendix

- It is strange to read this in the conclusions: "Note that also the presence of non-dust particles in the PM10 observations limit the assimilation accuracy; removal of the non-dust part as in (Jin et al., 2019a) should become part of the standard procedure." The first author of the paper is the first author of the referred paper. Why not removing the non-dust part in this paper?

References use here and not cited in the manuscript: Ginoux, P., J. M. Prospero, T. E. Gill, N. C. Hsu, and M. Zhao (2012), Global-scale attribution of anthropogenic and natural dust sources and their emission rates based on MODIS Deep Blue aerosol products, Rev. Geophys., 50, RG3005, doi:10.1029/2012RG000388.

---

## Referee Comment (RC2) · Anonymous Referee #3 · 16 Sep 2020

Research article review: acp-2020-435 "Source backtracking for dust storm inversion using adjoint method: case study of northeast China"

The manuscript proves very important issue related to missing sources in numerical atmospheric-dust models, which directly, due to the very high sensitivity of models to input dust source information, impact dust forecast quality. Horqin desert is in some literature recognized as a potential source area, but this information (as many other sources) is not included in many numerical atmospheric-dust forecast models. In Ginoux et al. (2012) and the report "Global Assessment of Sand and Dust Storms" (2016) this area is mentioned, but in terms of anthropogenic-dust sources. Horquin desert is

named as "Horquin sandy land". Sandy soils are not efficient dust emissive areas because of the relatively coarse soil texture, but in the reference is stated that this area contains anthropogenic dust sources. In the same reference the mean number of days with dust (M-DB2 DOD>0.2, for 2002-2009) for this region is relatively low (compared with emission frequency from major global sources) and appears in March-May and September-November seasons. Also, anthropogenic impact on increasing souce activity could be increasing in recent years. Since this manuscript proves significant emission originate from Horqin area, it proves the need for update of dust source information in models, or specific model used in the manuscript (LOTOS-EUROS).

This referee suggests that manuscript should be published after minor revision, as suggested bellow. It includes contribution for better understanding of east Asia dust source regions, usually underestimated in dust forecast models. The authors present alternative way to explore new dust source areas from airborne dust observation.

As the authors adopted suggested corrections by this referee in first cycle of review (before public discussion), only few more following suggestions are required form the authors to consider:

• General comment:

The authors need to explore published papers on dust sources and dust modeling that cover domain of northeast China, and include related findings – to emphasize the significance of the work presented in this manuscript and the necessity of upgrading dust source information in numerical models in order to reduce dust forecast uncertainties in this area. Few suggestions for such references are listed bellow.

Some of the references that recognized dust emission activity in this region are:

UNEP, WMO, UNCCD: Global Assessment of Sand and Dust Storms, United Nations Environment Programme, 2016., which includes already discussed: Ginoux, P., Prospero, M.J. Gill, T.E., Hsu, C. and Zhao, M.: Global scale attribution of anthropogenic

and natural dust sources and their emission rates based on MODIS Deep Blue aerosol products. Reviews of Geophysics, 50, RG3005, doi: 10.1029/2012RG000388, 2012., and DTF: Integrated Desert Terrain Forecasting for Military Operations, Earth and Ecosystem Sciences, 2013.

Reference about need for updating dust source masks in numerical dust forecast model: Kim, D., M. Chin, H. Bian, Q. Tan, M. E. Brown, T. Zheng, R. You, T. Diehl, P. Ginoux, and T. Kucsera, The effect of the dynamic surface bareness on dust source function, emission, and distribution, J. Geophys. Res., 118, doi: 10.1029/2012JD017907, 2013.

Reference that describe dust model simulation for this region, but with special intervention to include Horquin source: Zhang, DF, et al., Effects of climate changes on dust aerosol over East Asia from RegCM3, Advances in Climate Change Research, Volume 7, Issue 3, pp. 145-153, https://doi.org/10.1016/j.accre.2016.07.001

etc.

 c Technical comments:

Page 7, line 21:

Rephrase the sentence: "Note that the dust emission model computes hourly emissions per grid cell, which may vary strongly from hour to hour." to "Note that the dust emission model output data are on every hour per grid cell, and results may vary strongly from hour to hour. Also, dust concentration extremes that last less than one hour, can be missed in model output data. "

explanation (if I have understood well meaning of these data in the manuscript): numerical dust atmospheric model can calculate in time steps much lesser than one hour (depending on spatial resolution and setup of model for physics time steps), but model output data are what matters in presenting forecast and perform model verification. As is understood from the text, this sentence refers to the frequency of model output data.

Page 17, line 22-23:

At the end of paragraph, add the comment with reference (it considers the detection of high latitude sources): These emissions may originate from high-latitude sources, as one discussed in Bullard et al. (2016). (or rephrase it as you wish)

Bullard, J. E. et al. High-latitude dust in the Earth system. Rev. Geophys. 54, 447–485, 2016.

---

## Author Comment (AC1) · 20 Oct 2020

**Response to Referee #4:** We would like to thank the referee for the careful review and the thoughtful comments, which help us to further improve the quality of the manuscript.

Our response follows (*the reviewer's comments are in italics and blue*)

*Major Comments*

*Question 1) I find that the identification of the Horqin desert as the source that would explain the unresolved dust loads is not robust enough and would need further clarification or at least nuancing. There are few issues that make this identification uncertain:*
*a: The study uses Himawari-8 AOD. How can it be concluded that the AOD signals in regions MR-A and MR-B are due mostly to dust? The authors should provide evidence of that. Forecasts from the US Navy on May 3, 2017*
*(https://www.nrlmry.navy.mil/aerosol/globaer/ops_01/mongolia/201705/2017050312_globaer_ops_mongolia.gif) show a strong influence of sulfate optical depth in addition to dust. This effect could be strongly biasing the results as it would be overestimating the dust optical depth.*

**Reply:** Thanks for pointing out this issue. The bias in both AOD and PM10 of course would influence the model calibration as well as the assimilation, especially when the non-dust aerosol fraction (from anthropogenic activities) is relatively large. For example, the 2015 dust event that was studied in our other paper (Jin et al 2019a), where the max dust concentration is around one hundred to several hundred ug/m3 hence the non-dust part (around 100 ug/m3) is not negligible. However, the tested 2017 dust event in this paper is an extremely severe one when dust aerosol is very dominant, more information is added in the context. Therefore, our previous study (Jin et al., 2019b) and other related study (Mahowald et al., 2017) directly used AODs and PM observations to represent the dust loading during the severe 2017 dust event.

Besides, a variable observation representation error that developed in Jin et al. (2018) is used in this study to reduce the influence caused by the observation bias.

To make it clear, remarks are added in page 5, line 30-31 and page 7, line 1-8, by saying **"Though both the aforementioned Himawari-8 and the ground PM10 data are actually a sum of the dust aerosols and particles released in local activities, the 2017 dust storm events were reported as extremely severe ones with dust concentrations at downwind cities reaching up to 4,000 ug/m3; hence, dust aerosols are very dominant in the full aerosols. For such kinds of severe dust events, Mahowald et al. (2017) indicated AOD can be directly used as a reliable tool to represent dust loading in the atmosphere. AODs from Moderate Resolution Imaging**

*Spectroradiometer (MODIS) satellite products and the ground PM10 observations were also directly used to represent dust intensity for the same event in Zhang et al. (2018). Therefore, all these measurements are assumed to be representative for comparison with the dust simulations in this study too. In case of less severe dust storms, observational bias corrections (Dee and Uppala, 2009) would be required to remove the non-dust part from the observations to allow comparison with a 'dust-only' model (Jin et al., 2019a). Besides, a variable observation representation error is used in this study to reduce the observation bias influence as will be explained in Section. 6.2."*

*"A variable representation error designed in Jin et al. (2018) is used to represent the uncertainty of PM10 measurements, in which a smaller representation error is assigned to measurements reporting a higher PM10 value so that those small-value PM10 observations with larger non-dust fraction will has less influence in assimilation than those high-value measurements. "* in page 20, line 23-26.

Concerning the potential strong sulfate aerosol level over the northeast China indicated by the US Navy forecast, we have checked the SO2 observations which is an indicator of the sulfate aerosols. The daily (2017 May 03), monthly (2017 May) and yearly average SO2 concentration observations (2017) can be found as follows. The monthly and yearly average are treated as the normal levels here. Actually, the SO2 concentrations over the northeast China are relatively low (or even lower) compared to the normal values (2017 May monthly or 2017 Annually). Therefore, the simulated sulfate AOD plume is a negligible one during this event.

[Figure]

*Daily average SO2 concentration observation (2017 May 03)*

[Figure]

*Monthly average SO2 concentration observation (2017 May)*

[Figure]

*Yearly average SO2 concentration observation (2017)*

*b: Figure 8 shows the backward time series of emission sensitivity. The conclusion that the Horqin desert is the most likely source is based on an a priori assumption: (Page 17, line 5; "If the dust was emitted earlier, it seems to originate from regions further south. However, these are densely populated regions covered with vegetation, and therefore not a likely to be a source of dust.") There are two problems with this statement: 1) The areas highlighted in panels a.5 and a.6 overlap with potential dust emission areas highlighted in the in-depth study and dust source inventory of Ginoux et al. (2012) (see their Figure 10, region #6, North China Plains). 2) If the air masses are coming from the southern populated areas they may come with anthropogenic aerosols (this is consistent with the trajectory of the high sulfate optical depth from the Navy forecasts highlighted above) which goes back to my previous comment.*

**Reply**:Yes, the further south region (North China Plains) is a potential aerosol/dust source. However, it is also not the case for this event since the ground-based PM10 observations report quite normal low values (no dust) on the passage from the North China Plain to marked region (MR-A). To make it clear, remarks are added by saying

"*Similar conclusions were drawn for the severe dust event ("SD3"), for which figures of backward emission sensitivities are available as supplementary material. For SD3, it was noticed that dust emissions from the Horqin desert between May 06 09:00 to 15:00 could explain the high dust loads observed. If the dust was emitted earlier, it seems to originate from further south regions, mainly the north China plain. Though it is also considered as a weak potential source by Ginoux et al. (2012), it is a densely populated region covered with vegetation, and therefore treated to be further less likely to be a source of dust in our model. Besides, the PM10 observations at May 03 03:00 and 06:00 corresponding to the last two snapshots in Fig.8(a) are presented in Fig.S3 in the Supplementary Material, they show that the north China plains were clear of dusts during the event.*" in page 18, line 25-32.

[Figure]

***Figure 8. Backward time series of emission sensitivity of the dust simulation at MR-A_6 2017 May 03, 19:00 CST: emission sensitivity distribution at 2017 May 03, 18:00 (a.1), 15:00 (a.2), 12:00 (a.3), 09:00 (a.4), 06:00 (a.5), 03:00 (a.6).***

[Figure]

***Figure S3.PM10 observations at May 03 03:00 CST (a), May 03 06:00 (b)***

*c. On a side note the Ginoux et al. (2012) inventory already highlights the Horqin desert as a potential dust source, and it is not the only study. Please note that the absence of emissions in your model there may be model-specific. Many models do not assume zero emissions from sparsely vegetated areas and use LAI or fractional cover to scale emissions in those areas so they are not zero.*

**Reply**: Accepted. More information regarding the Horqin desert is given in page 4, line 18-25 "***Horqin desert which is also named as Horqin sandy land is mixed with sparse vegetation, agriculture lands in northeastern China. Though it is recognized as one potential emission source in several dust models but is also considered of far less importance compared to other major ones, e.g., Gobi and Mongolia desert, and Taklamakan desert (Zhang et al., 2003, Ginoux et al., 2012, UNEP. Et al., 2016). Kim et. al. (2013) suggested a dynamic vegetation index is essential for representing the seasonal bareness variation that regulates dust emissions over this region. Zhang et al. (2016) predicted a declining trend in dust emission from this sandy land due to the climate change. For LOTOS-EUROS used in this work and another model BSC-DREAM8b, it is not present as an easily erodible in the dust emission scheme at least for these tested severe events.***"

Based on our results, we add the suggestion that "***Based on our results, it is advised that dust sources in dust transport models include Horqin desert as a more active source region, especially when severe dust is observed in the northeast China.***" in page 24, line 13-14.

*2) The comparison of the final results with those from the previous study (Jin et al. 2019b) after including the new source in the model is not consistent. At least that is what I understood from reading the paper. In Jin et al. (2019b) AOD is assimilated from the Himawari-8 satellite AOD and PM10 is used as an independent evalua- ̆tion dataset. In this paper, PM10 is also used in the assimilation (page 19, line 15: "Himawari-8 AOD values are assimilated in the first cycle, of which the measurement error configurations are similar as in Jin et al. (2019b). However, almost no AOD values are retrieved in the second window over the MR-A region, hence the ground PM10 observation are assimilated instead, of which the representation errors are set similar to those in Jin et al. (2019a)."). Therefore, the comparison between both experiments is inconsistent and the improvements (or at least a large fraction of them) stated in the text and the abstract haven't been shown to be due (at least solely) to the inclusion of the new dust source, and they could just be due to the assimilation of the PM10 (which is presumably used as the evaluation*

*dataset as well). Experiments should be compared under the same assimilation conditions both without and with PM10.*

**Reply:** Thanks for point out the issue. It is indeed unfair to compare the emission inversion in previous study and the improved one, since both new observations (PM10) and improved emission models are introduced together. Therefore, an extra experiment is now added, and remarks are added in by saying

**"*It is actually unfair to compare the results between the emission inversion in Jin et al. (2019b) and the proposed one using improved emission model directly, since the difference could be caused either by extra PM10 observations or by the emission schemes introduced. Therefore, an extra emission inversion is conducted. As shown in Table. 2, it repeats the previous emission inversion exactly but extra PM10 observations over the MR-A region are assimilated. The extra emission inversion results in the consistent posterior to the emission inversion in Jin et al. (2019b). The posterior simulation of extra emission inversion is not shown in this paper, but the RMSE time series over MR-A region is calculated and given in Fig. 7(b) which shows relative same to the posterior of emission inversion in Jin et al (2019b). It is because the existing emission background error covariance that explains the emission spread cannot resolve extra PM10 measurements. The comparison indicates that solely assimilating extra PM10 but without using improved emission modeling has no effect on improving dust emission inversion over the northeast China.***"** in page 20, line 32-33 and page 21, line 1-3.

[Figure]

*Figure 7. a: Hourly PM10 observations averaged over MR-A (13 sites), with shaded area indicating the standard deviation range; b: root mean square error of prior and posterior simulation, posterior of extra emission inversion, reconstructed prior and posterior within MR-A. The location of the stations is indicated in Fig. 2.*

*Table 2. Dust storm emission inversions*

|  | 1st analysis assimilated data | 2nd analysis assimilated data |
|---|---|---|
| *prior* | - | - |
| *emis inver* | Himawari-8 AOD | Himawari-8 AOD |
| extra *emis inver* | Himawari-8 AOD and $PM_{10}$ | Himawiri-8 and and $PM_{10}$ |
| *prior* using improved emis | - | - |
| *emis inver* using improved emis | Himawari-8 AOD | Himawari-8 AOD and $PM_{10}$ |

Note that -: no assimilation; *emis inver*: emission inversion; improved emis: improved emission model.

*3) Structure of the paper: I strongly recommend restructuring the paper to follow the classic introduction, data and methods, results, discussion, conclusion format. The paper combines methods, data with results and discussion thought-out the paper and this makes the reading difficult.*

**Reply:** The structure of the paper is re-organized to make the reading easier. Sections listed are now "***Introduction***", "***Measurements***", "***Dust model***", "***Adjoint model***", "***Dust emission inversion***", "***Result and discussion***", "***Summary and conclusion***".

*Additional comments or minor issues:*

*Equation 3 in the original publication of Ginoux et al. (2001) is the same expression but to the power of 5.*

**Reply**: This is a typo error not only in this paper but also in the first author's thesis. What we used is exactly same to the one in the original publication (Ginoux et al. 2001). Now it is corrected in the manuscript, and a statement is also made by saying "***Note that a typo error was made in the same terrain preference equation but without a power of 5 in the related Ph.D. thesis (Jin, 2019).***" in page 8, line 16-17.

*Page 7, line 32: there is a typo ("eThe red box")*

**Reply**: Corrected.

*- Again, I recommend restructuring the paper and include all the methods after the introduction. Also in my opinion, section 5.2 interrupts the flow of the manuscript. It would be enough to say that Âaboth the finite difference and adjoint method seem able ˘ to derive emission sensitivities and refer to an include this section in an appendix*

**Reply**: We agree with the referee. Now **measurements**, systems/methodology sections including **dust model, adjoint model**, and **dust emission inversion** are introducing after the **Introduction**.

*It is strange to read this in the conclusions: "Note that also the presence of non-dust particles in the PM10 observations limit the assimilation accuracy; removal of the nondust part as in (Jin et al., 2019a) should become part of the standard procedure." The first author of the paper is the first author of the referred paper. Why not removing the non-dust part in this paper?*

**Reply**: Following the reply to **General comments 1**, the dust bias correction is more important when the dust aerosols are not so dominant, such as the 2015 dust case studied in our previous study (Jin et al., 2018). In that study, deep learning model training was performed over every monitoring station to simulate the dynamic non-dust aerosol. The non-dust aerosol removal is not so necessary for this severe 2017 case but would require a huge amount of energy for deep learning training. Remarks are supplemented in page 24, line 9-11"***Note that also the presence of non-dust particles in the PM10 observations limit the assimilation accuracy; removal of the nondust part as in (Jin et al., 2019a) should become part of the standard procedure.***" is now changed to "***Note that also the presence of non-dust particles in the PM10 observations limits the assimilation accuracy; removal of the non-dust part as in (Jin et al.,2019a) will become part of the standard procedure in the future when the required huge amount of computing power is available.***"

*References use here and not cited in the manuscript: Ginoux, P., J. M. Prospero, T. E. Gill, N. C. Hsu, and M. Zhao (2012), Global-scale attribution of anthropogenic and natural dust sources and their emission rates based on MODIS Deep Blue aerosol products, Rev. Geophys., 50, RG3005, doi:10.1029/2012RG000388.*

**Reply**: This is of course an important reference related to our study, and it is referred now.
"***Due to the growing interest in dust storms, the understanding of the physical processes associated with the dust cycles has increased rapidly over the last decades (Ginoux et al., 2012; World Meteorological Organization, 2018).***" in page 2, line 10-11.

*"Horqin desert which is also named as Horqin sandy land is mixed with sparse vegetation, agriculture lands in northeastern China. Though it is recognized as one potential emission source in several dust models but is also considered of far less importance compared to other major ones, e.g., Gobi and Mongolia desert, and Taklamakan desert (Zhang et al., 2003; Ginoux et al., 2012; UNEP. et al., 2016)."* in page 4, line 19-21.

---

## Author Comment (AC2) · 20 Oct 2020

**Response to Referee #3:** We would like to thank the referee for the careful review and the insightful comments, especially the reference source concerning the dust emission over Horqin desert, which help us to further improve the quality of the manuscript.

Our response follows (*the reviewer's comments are in italics and blue*)

*General comments*

*The authors need to explore published papers on dust sources and dust modeling that cover domain of northeast China, and include related findings – to emphasize the significance of the work presented in this manuscript and the necessity of upgrading dust source information in numerical models in order to reduce dust forecast uncertainties in this area. Few suggestions for such references are listed bellow*

*Some of the references that recognized dust emission activity in this region are: UNEP, WMO, UNCCD: Global Assessment of Sand and Dust Storms, United Nations Environment Programme, 2016., which includes already discussed: Ginoux, P., Prospero, M.J. Gill, T.E., Hsu, C. and Zhao, M.: Global scale attribution of anthropogenic and natural dust sources and their emission rates based on MODIS Deep Blue aerosol products. Reviews of Geophysics, 50, RG3005, doi: 10.1029/2012RG000388, 2012., and DTF: Integrated Desert Terrain Forecasting for Military Operations, Earth and Ecosystem Sciences, 2013.*

*Reference about need for updating dust source masks in numerical dust forecast model: Kim, D., M. Chin, H. Bian, Q. Tan, M. E. Brown, T. Zheng, R. You, T. Diehl, P. Ginoux, and T. Kucsera, The effect of the dynamic surface bareness on dust source function, emission, and distribution, J. Geophys. Res., 118, doi: 10.1029/2012JD017907, 2013.*

*Reference that describe dust model simulation for this region, but with special intervention to include Horquin source: Zhang, DF, et al., Effects of climate changes on dust aerosol over East Asia from RegCM3, Advances in Climate Change Research, Volume 7, Issue 3, pp. 145-153, https://doi.org/10.1016/j.accre.2016.07.001*

**Reply**: Thanks for providing the rich reference source concerning the Horin desert dust emission. Discussion about the dust emission over the Horqin desert in the northeast China is supplemented in page 4, line 18-23. "***Horqin desert which is also named as Horqin sandy land is mixed with sparse vegetation, agriculture lands in northeastern China. Though it is recognized as one potential emission source***

*in several dust models but is also considered of far less importance compared to other major ones, e.g., Gobi and Mongolia desert, and Taklamakan desert (Zhang et al., 2003; Ginoux et al., 2012; UNEP. Et al., 2016). Kim et al. (2013) suggested a dynamic vegetation index is essential for representing the seasonal bareness variation that regulates dust emissions over this region. Zhang et al. (2016) predicted a declining trend in dust emission from this sandy land due to the climate change.*"

*Technical comments:*

*Page 7, line 21: Rephrase the sentence: "Note that the dust emission model computes hourly emissions per grid cell, which may vary strongly from hour to hour." to "Note that the dust emission model output data are on every hour per grid cell, and results may vary strongly from hour to hour. Also, dust concentration extremes that last less than one hour, can be missed in model output data. " explanation (if I have understood well meaning of these data in the manuscript): numerical dust atmospheric model can calculate in time steps much lesser than one hour (depending on spatial resolution and setup of model for physics time steps), but model output data are what matters in presenting forecast and perform model verification. As is understood from the text, this sentence refers to the frequency of model output data.*

**Reply**: Accepted.

*Page 17, line 22-23: At the end of paragraph, add the comment with reference (it considers the detection of high latitude sources): These emissions may originate from high-latitude sources, as one discussed in Bullard et al. (2016). (or rephrase it as you wish) Bullard, J. E. et al. High-latitude dust in the Earth system. Rev. Geophys. 54, 447–485, 2016.*

**Reply**: Yes, potential dust source from the high-latitude is now mentioned in page 18, line 27-28 "***Earlier emissions are traced northwards from regions in Siberia or other high-latitude regions as discussed in Bullard et al. (2016) that are still not identified as active source in dust emission models.***"